# MicroRNAs as Biomarkers and Therapeutic Targets in Inflammation- and Ischemia-Reperfusion-Related Acute Renal Injury

**DOI:** 10.3390/ijms21186738

**Published:** 2020-09-14

**Authors:** Yueh-Lin Wu, Hsiao-Fen Li, Hsi-Hsien Chen, Heng Lin

**Affiliations:** 1Graduate Institute of Clinical Medicine, College of Medicine, Taipei Medical University, Taipei 110, Taiwan; vincewu168@gmail.com; 2Division of Nephrology, Department of Internal Medicine, School of Medicine, College of Medicine, Taipei Medical University, Taipei 110, Taiwan; 3Division of Nephrology, Department of Internal Medicine, Taipei Medical University Hospital, Taipei 110, Taiwan; 4TMU Research Center of Urology and Kidney, Taipei Medical University, Taipei 110, Taiwan; 5Department of Physiology, School of Medicine, College of Medicine, Taipei Medical University, Taipei 110, Taiwan; bubble0728@gmail.com; 6PhD Program in Biotechnology Research and Development, College of Pharmacy, Taipei Medical University, Taipei 110, Taiwan

**Keywords:** microRNA, acute kidney injury, biomarker

## Abstract

Acute kidney injury (AKI), caused mainly by ischemia-reperfusion, sepsis, or nephrotoxins (such as contrast medium), is identified by an abrupt decline in kidney function and is associated with high morbidity and mortality. Despite decades of efforts, the pathogenesis of AKI remains poorly understood, and effective therapies are lacking. MicroRNAs (miRNAs) are small noncoding RNAs that regulate gene expression at the posttranscriptional level to control cell differentiation, development, and homeostasis. Additionally, extracellular miRNAs might mediate cell–cell communication during various physiological and pathological processes. Recently, mounting evidence indicates that miRNAs play a role in the pathogenesis of AKI. Moreover, emerging research suggests that because of their remarkable stability in body fluids, microRNAs can potentially serve as novel diagnostic biomarkers of AKI. Of note, our previous finding that miR-494 is rapidly elevated in urine but not in serum provides insight into the ultimate role of urine miRNAs in AKI. Additionally, exosomal miRNAs derived from stem cells, known as the stem cell secretome, might be a potential innovative therapeutic strategy for AKI. This review aims to provide new data obtained in this field of research. It is hoped that new studies on this topic will not only generate new insights into the pathophysiology of urine miRNAs in AKI but also might lead to the precise management of this fatal disease.

## 1. Introduction

Acute kidney injury (AKI) is a clinical syndrome characterized by sharply decreased renal function that is complicated by metabolite retention and water, electrolyte, and acid–base imbalance. Importantly, the sequelae of AKI are quite severe and critical, such as increasing the risk of in-hospital mortality, the incidence of chronic kidney disease (CKD), and the progression to end-stage renal disease. Currently, no effective treatments for AKI are available except for supportive care in the practice of dialysis. Therefore, the early detection of AKI, the identification of the risks of AKI, and raising awareness of AKI have become a global issue [1]. MicroRNAs (miRNAs) are small noncoding RNAs of approximately 19–25 nucleotides in length that are involved in the regulation of gene RNA expression at the posttranscriptional level [2]. Emerging evidence suggests that miRNAs play essential roles in the pathogenesis of AKI [3]. As a silver lining in modern nephrology, recent investigations of miRNAs in AKI may lead to the development of novel diagnostic tools and therapeutic interventions.

### 1.1. Definitions of AKI

The concept of clinical consequence after acute kidney insult named acute renal failure (ARF) used in the past placed excessive emphasis on whether renal function had overtly failed as determined by a bedside observation. However, recently, it is believed that the whole picture of AKI is a dynamic process that includes initiation, maintenance, and recovery phases, while further including clinical findings and laboratory investigation [4]. Therefore, by this new concept, the severity of AKI is broad, ranging from the minor increment in serum creatinine to overt kidney failure requiring renal replacement therapy. Regrettably, serum creatinine is not an ideal biomarker of AKI. Serum creatinine’s half-life may increase from 4 h to 24–72 h after a definite renal insult. The serum creatinine concentration might take 24–36 h to accumulate before being detected as abnormal. Thus, this definition criterion of AKI based only on the increase, even if minimal, of serum creatinine does not intercept early kidney damage that precedes the reduction of glomerular filtration rate.

### 1.2. Epidemiology of AKI

AKI is a common complication in patients hospitalized for acute illness. A systematic review of 312 cohort studies found that AKI occurred in one of five adults and one of three children hospitalized with acute illness [5]. Of note, despite modern medical facilities and better support, AKI-associated mortality remains unacceptably high. Recently, studies have shown that unadjusted mortality associated with an episode of AKI is estimated at 23.9% in adults and 13.8% in children [5]. Furthermore, the adjusted risk of in-hospital mortality has shown near-linear increases with worsening severity of AKI [6].

### 1.3. Causes of AKI

Clinically, the major causes of AKI include ischemia/reperfusion (I/R), sepsis, and various nephrotoxins, such as cisplatin and contrast medium. The pathogenesis of AKI is multifactorial, involving tubular injury, vascular dysfunction, and inflammation. Multiple cell types and different cellular processes and molecular mediators/regulators are responsible for the initiation and progression of the disease [7].

#### 1.3.1. AKI Induced by Ischemia/Reperfusion Injury

Despite research advances in the past few decades, the pathophysiology of AKI is still not fully understood. The classical ischemia/reperfusion model of AKI provides insight into the possible cellular and molecular mechanisms of AKI.

After the insult of ischemia-reperfusion injury, proximal tubular cells rapidly lose the integrity of their cell membrane and cytoskeleton. The alterations in the location of adhesion molecules and Na^+^/K^+^-ATPase results in the normally highly polar epithelial cell losing polarity. Alterations in the cytoskeleton regulate changes in cell polarity, cell–cell interactions, and cell-matrix interactions. With increasing time or severity of ischemia, cell death occurs by either necrosis or apoptosis. Inflammatory cells are further recruited to the injured area and release proinflammatory factors, chemokines, and costimulatory molecules, such as tumor necrosis factor-alpha (TNF-α), interleukin-1 (IL-1), interleukin-6 (IL-6), and monocyte chemoattractant protein-1 (MCP-1) [7,8]. These molecules can not only worsen the damage to renal tubular epithelial cells and endothelial cells but can also further amplify the inflammatory response, promote inflammatory infiltration, and facilitate the “inflammatory cascade effect”. Finally, viable epithelial cells migrate and cover denuded areas of the basement membrane. The cells then undergo division and replace lost cells. Ultimately, the cells go on to differentiate and re-establish the normal polarity of the epithelium. Therefore, the pathophysiology of this injury represents a complex interplay between the vasculature, the tubules, and inflammation. Interestingly, both innate and acquired immunity mainly contribute to the initial injury phase, the regulation of the inflammatory response, and the repair phase [8].

#### 1.3.2. AKI Induced by Sepsis

Septic AKI has a pathophysiological mechanism that clearly differs from that of ischemic AKI [9,10] Alterations in renal blood flow, microcirculatory disturbances, and blood flow redistribution between the renal cortex and medulla are potentially important contributors to renal tubular injury. However, due to the differences between animal models and septic patients, there is limited evidence indicating the pathophysiological mechanism of septic AKI.

#### 1.3.3. Contrast-Induced Nephropathy

The pathophysiology of contrast-induced nephropathy (CIN) is complex and only partially understood. What exactly happens inside a human kidney in vivo can only be speculated from the results of mainly animal and laboratory studies. Hypoxic medullary injury plays a critical role in CIN [11,12]. This is caused by three different but potentially interacting pathways: hemodynamic effects of contrast medium, the effect of reactive oxygen species (ROS), and direct contrast medium-induced tubular cell toxicity.

## 2. MicroRNA Biogenesis and Function

Over the past decades, miRNAs have been shown to play a critical role in the regulation of almost all biological cell functions, including proliferation, differentiation, metabolism, and apoptosis [13]. Recently, emerging data further supported that miRNAs are involved in the pathogenesis of many human [14] diseases.

### 2.1. The Canonical Pathway of miRNA Biogenesis

The dominant pathway of the biogenesis of most miRNAs is the canonical pathway. In this pathway, miRNAs are transcribed from DNA sequences into primary miRNAs (pri-miRNAs), which contain approximately 300–1000 nucleotides. After its production, a pri-miRNA is further cleaved into precursor miRNA (pre-miRNA) at its stem-loop structure by the RNase III enzyme Drosha [15] and its cofactor DGCR8 (DiGeorge syndrome critical region gene 8 or Pasha) [16], which is composed of approximately 70 nucleotides [17]. The pre-miRNA is then transported from the nucleus into the cytoplasm by an exportin 5/ GTP-binding nuclear protein Ran (RanGTP) complex [18] and is then further cleaved by the RNase III enzyme Dicer to yield a single-stranded mature miRNA [19]. The name of the mature miRNA form is determined by the directionality of the miRNA. The 5p strand arises from the 5′ end of the pre-miRNA hairpin, while the 3p strand originates from the 3′ end. To perform its function, a miRNA interacts with the argonaute (AGO) protein to form an effector complex called the RNA-induced silencing complex (RISC). RISC binds to the 3′-untranslated region (UTR) of a target messenger RNA (mRNA), leading to the repression of protein translation or mRNA degradation. Of note, evidence suggests that a miRNA binds to a specific “seed sequence” to regulate its target gene(s), resulting in either translational repression, mRNA deadenylation, and decapping by 3′ UTR binding [20], or silencing by 5′ UTR binding [16]. Furthermore, in most conditions, the complementarity between mammalian miRNAs and mRNA targets is not fully complete [21]. Perfectly complementary target mRNAs are mainly cleaved by protein argonaute-2 (AGO2) in humans [22]. In cases of partial complementarity, AGO proteins recruit the GW182 family to mediate gene silencing [20,23]. GW182 proteins play a central part in this silencing process and function as flexible scaffolds to bridge the interaction between AGO proteins and downstream effector complexes [2].

### 2.2. The Noncanonical Pathways of miRNA Biogenesis

In addition to the canonical miRNA biogenesis pathways, multiple noncanonical miRNA biogenesis pathways have been proposed. These pathways utilize different combinations of the main proteins, such as Drosha, Dicer, exportin 5, and AGO2, which are involved in the canonical pathway. In general, noncanonical miRNA biogenesis can be grouped into Drosha/DGCR8-independent and Dicer-independent pathways.

#### 2.2.1. Drosha/DGCR8-Independent Pathway

A noncanonical pathway was first described to take place during mirtron production, in which the Drosha-mediated processing step is bypassed and pre-miRNA is produced from mRNA introns during splicing [24]. Another example is the 7-methylguanosine (m^7^G)-capped pre-miRNAs, which are endogenous short hairpin RNAs generated directly through transcription (for example, miR-320). These nascent RNAs are then directly exported to the cytoplasm through exportin 1 without the need for Drosha cleavage. Following Dicer processing, the 3p miRNA is strongly selected, most likely due to the m^7^G cap preventing 5p miRNA loading onto AGO [25].

#### 2.2.2. Dicer-Independent Pathway

Dicer-independent miRNAs are processed by Drosha from endogenous short hairpin RNA (shRNA) transcripts, and one example miR-451 [26]. Drosha generates a short pre-miR-451 hairpin with a stem of ~18 bp that is too short to be processed by Dicer [26]. Thus, pre-miR-451 is directly loaded onto AGO2 and cleaved by the AGO catalytic center to generate an intermediate 3p strand, which is then further trimmed [27].

## 3. Extracellular microRNAs in Biological Fluids

Many studies have shown that miRNAs can be detected in human biological fluids, such as plasma [28], serum [29], urine [30], cerebrospinal fluid [31], saliva [32], breast milk [33], bronchial lavage, tears, colostrum, peritoneal fluid, and seminal fluid [34]. One of the main features of extracellular miRNAs is their high stability in body fluids, even under extreme conditions, such as high RNase activity, low or high pH, long-term room temperature storage, or multiple freeze–thaw cycles [28]. For example, in human urine samples, even after seven cycles of freezing and thawing or 72 hour-long storage at room temperature, miRNA levels remained unchanged [35]. More importantly, similar to intracellular miRNAs, extracellular miRNAs have been suggested to be involved in cell–cell communication and play a regulatory role in physiological and pathological processes [36]. Extracellular miRNAs are released via three different pathways: (1) active transport via extracellular vesicles (EVs), (2) active secretion via an RNA-binding protein-dependent pathway, and (3) passive leakage from broken or damaged cells.

### 3.1. Extracellular microRNA Transport via Extracellular Vesicles

EVs are membrane-bound vesicles released by cells for intercellular communication. In general, smaller EVs (<200 nm) generated by exocytosis of intraluminal vesicles through the fusion of multivesicular bodies (MBs) with the plasma membrane are called exosomes, while larger EVs (>200 nm) formed by direct shedding, outward budding and fission of the plasma membrane are called microvesicles (Some authors suggested also including intermediate vesicles). Both exosomes and microvesicles can serve as mechanisms of miRNA release into the circulation of extracellular fluid.

Even with many research efforts over the years, the miRNA secretory mechanism is still obscured. Kosaka et al. found that miRNAs are released through ceramide-dependent secretory machinery [37] but not the endosomal sorting complex required for transport (ESCRT) system, which is essential for exosomes being targeted to lysosomes [38]. Neutral sphingomyelinase 2 (nSMase2) has been shown to regulate ceramide biosynthesis and act as the key molecule in the secretion of exosomal miRNAs [37].

Growing evidence has shown that cells might preferentially select particular miRNA populations and sort them into EVs [39]. However, the mechanism by which the miRNA content of EVs is determined is a critical question and still poorly understood. Some studies have suggested a role for AGO2 [40] and other RNA-binding proteins, such as heterogeneous nuclear ribonucleoprotein A2B1 (hnRNPA2B1) [41] and Y-box protein 1 [42], in the regulation of miRNA loading into exosomes.

### 3.2. Extracellular microRNA Transport via RNA-Binding Proteins

Several studies have shown that a handful of miRNAs are still detectable in the microvesicle-free fraction after isolation of exosomes/microvesicles using high-speed ultracentrifugation from biofluid [43,44], indicating the presence of non-vesicle-associated miRNAs.

Vickers et al. have reported that high-density lipoprotein (HDL) can transport endogenous miRNAs in the circulation and suggest that through divalent cation bridging, HDL could bind to extracellular plasma miRNAs and possibly protect miRNAs from external RNases [45]. Furthermore, the bound miRNAs can be taken up via class B type I scavenger receptors to then promote target mRNA inhibition in recipient cells [45]. Notably, the authors also demonstrated that the inhibition of nSMase2 significantly increased the amount of a specific miRNA exported to HDL, while previous data showed that the overexpression of nSMase2 promoted the release of exosomal miRNAs [45]. These results suggest that the export of specific miRNAs through the exosomal pathway and the HDL-dependent pathway may be opposing mechanisms, although nSMase2 activity is involved in both pathways.

In mammals, AGO2 is one of the major components of the RISC complex.

In 2011, two independent research groups demonstrated that most circulating extracellular miRNAs are not associated with exosomes but bound to AGO2 protein [46,47]. In size-exclusion chromatography studies, most plasma miRNAs were copurified with the AGO2 ribonucleoprotein complex, but only a few miRNAs, such as miR-16 and miR-92a, were predominantly associated with vesicles [47]. Another RNA-binding protein, nucleolar protein nucleophosmin 1 (NPM1), has been found to carry miRNA and may play a role in protecting miRNA from degradation [48]. Although these data showed that large portions of the circulating extracellular miRNAs are vesicle-free, protein-bound miRNAs, these miRNAs may be byproducts of cell death [46].

## 4. miRNAs in Acute Kidney Injury

As miRNAs are critical regulators of the fundamental activities of a cell from proliferation and differentiation to apoptosis, the number of studies on the functions of miRNAs in AKI progression is rapidly growing. The first evidence of miRNAs having pathological roles in AKI was established by a conditional knockout mouse model in which Dicer was ablated specifically from kidney proximal tubule (PT) [49]. In this model >80% of miRNAs were depleted from the renal cortex, and the PT-Dicer^−/−^ mice showed remarkable resistance to ischemic AKI, with significantly better renal function, less tissue damage, less tubular apoptosis, and better survival than wild-type mice after I/R injury [49]. These observations suggested a critical role for Dicer and associated miRNA production with the pathogenesis of ischemic AKI.

### 4.1. miR-687

The same team also examined microRNA expression by microarray analysis in the I/R mouse model and found that 13 specific miRNAs were increased, while a dozen of miRNAs were decreased [49]. Of note, among the increased miRNAs, miR-687 was markedly upregulated (>1700-fold) in mouse renal cortical tissue after I/R injury and in cultured kidney cells during hypoxia. The induction of miR-687 was decreased in hypoxia-inducible factor-1 (HIF-1) knockout cells and proximal tubule-specific HIF-1 knockout mice, which suggested that HIF-1 induced miR-687 [50]. Further bioinformatics analysis indicated phosphatase and tensin homolog (PTEN) as a target of miR-687. Blocking miR-687 preserved PTEN expression and attenuated cell cycle activation and renal apoptosis, resulting in protection against ischemic AKI in mice. Collectively, these results reveal that miR-687 plays a pathogenic role in ischemic AKI and demonstrate a novel HIF-1/miR-687/PTEN signaling pathway in I/R injury that may be targeted for therapy [50] 

### 4.2. miR-24

miR-24 has been shown to be upregulated in the kidney after I/R injury in mice, and silencing of miR-24 in vivo ameliorated renal I/R injury, inflammation, and renal function as well as overall survival in vivo [51]. Overexpression of miR-24 enhanced proximal tubular and endothelial cell apoptosis by downregulating sphingosine-1-phosphate receptor 1 (S1PR1), H2A histone family, member X (H2A.X), and heme oxygenase-1 (HO-1) protein [51]. H2A.X and HO-1 protect against DNA damage and oxidative stress [52,53], while S1P1R agonists directly block apoptosis and induce cell survival pathways via activation of the Akt and/or mitogen-activated protein kinase (MAPK) pathways [54]. Of note, postischemic fibrosis development was also highly attenuated in mice after miR-24 inhibition, which might be attributed to enhanced capillary density and tubular epithelial cell survival. Finally, miR-24 was upregulated in the kidneys of renal transplant recipients with prolonged cold ischemia time, indicating its potential role in human renal I/R injury [51].

### 4.3. miR-494

Our group found that miR-494 was upregulated quickly in kidney tissues after I/R injury and targeted activating transcription factor 3 (ATF3). In vivo, overexpression of miR-494 not only induced the expression of inflammatory mediators, such as IL-6, MCP-1, and P-selectin, but also promoted the NF-kB-dependent inflammatory response, resulting in exacerbated apoptosis and a further decrease in renal function [55]. Of note, we also found that the elevated expression of urinary miR-494 preceded the expression of serum creatinine in mice after I/R injury. Furthermore, there were no differences in the serum concentrations of miR-494, suggesting its renal or urinary tract origin. The delivery of antisense-miR-494 to mice before I/R injury rescued renal function, reduced proinflammatory cytokines, and caused suppression of caspase-3 apoptotic activity [55]. Accordingly, the serum miR-494 levels in healthy volunteers (healthy control), intensive care unit (ICU) patients without AKI, and ICU patients with AKI did not differ significantly, while the urinary levels of miR-494 in ICU patients suffering from AKI were significantly higher compared with those in the healthy control group [55].

Since the previous finding of the negative regulation between miR-494 and ATF3, we further isolated and measured the exosomal ATF3 mRNA levels in these urine samples. The ATF3 mRNA level in urinary extracellular vesicles was also higher in patients with AKI in the ICU compared with levels in healthy individuals [56]. Of note, total urinary ATF3 mRNA levels were not different. In a mouse model of AKI, we also found that the urinary exosomal ATF3 mRNA level was increased significantly as early as 1 h after I/R, with a simultaneous increase in the ATF3 mRNA level in the kidney tissue [56]. Previously, Yoshida et al. showed that overexpression of ATF3 in mice via adenovirus-mediated gene transfer ameliorated I/R injury, and our group found that ATF3 knockout (ATF3 KO) mice had higher renal I/R-induced mortality [57] and kidney dysfunction compared with wild-type mice [58]. We then asked whether the epithelium-derived exosomal ATF3 mRNA had a protective role after I/R injury. First, epithelium-derived exosomal ATF3 mRNA decreased MCP-1 expression in epithelial cells and macrophage infiltration after hypoxia/reoxygenation (H/R) [56]. Second, we noted that knockdown of ATF3 increased I/R-induced MCP-1 mRNA expression both in vitro and in vivo [56]. Finally, and most interestingly, intrarenal pelvic injection of exosomal ATF3 ameliorated I/R-induced renal dysfunction in ATF3 KO mice compared to wild-type mice [56]. In contrast, the administration of exosomal Flag did not affect I/R-induced renal dysfunction [56]. These data suggest that the increased level of ATF3 mRNA in extracellular vesicles in the urine reflects a protective signaling effect through extracellular vesicle uptake and activation of ATF3-responsive gene expression programs in target cells.

### 4.4. miR-21

Although miR-21 has the most consistently reported association with AKI, the data are difficult to interpret, and the miRNA appears to play a dual role. It has been proposed that miR-21 is a two-edged sword in ischemic AKI: neither knockdown nor overexpression led to adverse consequences [59]. The deletion of miR-21 significantly promotes cell death [60], whereas overexpression of miR-21 suppresses cell death by downregulating programmed cell death protein 4 (PDCD4) in septic AKI [61]. Additionally, emergent data showed that I/R injury suppressed miR-21 expression in bone marrow-derived dendritic cells, and mice with miR-21 deficiency in dendritic cells aggregated I/R injury [62]. One clinical study found that miR-21 levels increased both in plasma and urine with increasing AKI severity after adult cardiac surgery [63]. However, another clinical study found decreased, but not increased, expression of miR-21 in cardiac surgery-associated AKI patients [64]. Recently, a large-scale study found that urine and blood miR-21 expression in the remote ischemic preconditioning group increased significantly in children with congenital heart disease undergoing cardiopulmonary bypass surgery [65]. These findings may reflect a pleiotropy of miR-21 targets with a complex regulatory network.

## 5. miRNAs as Novel Biomarkers for AKI

### 5.1. Circulating miRNAs

As miRNAs could be secreted into many different body fluids, such as serum and urine, they have been proposed to serve as novel biomarkers for the early diagnosis of AKI (Table 1). However, current human experiments are characterized by variability in the described microRNA responses and lack consistent results. This may be related to the different pathogenesis types in AKI or to a complex but rapidly responding miRNA-associated regulatory network. An early observation study showed that circulating levels of miR-16 and miR-320 were downregulated in the plasma of AKI patients, whereas miR-210 was upregulated compared with that in healthy controls [66]. The study also demonstrated that miR-210 could be a robust independent predictor of 28-day survival in critically ill patients with AKI [66]. A further study identified a set of 10 selected miRNAs (miR-101-3p, miR-127-3p, miR-210-3p, miR-126-3p, miR-26b-5p, miR-29a-3p, miR-146a-5p, miR-27a-3p, miR-93-3p and miR-10a-5p) for AKI diagnosis in ICU patients. This panel of miRNAs showed a high diagnostic value with nearly 100% sensitivity and specificity [67]. In the same study, another set of four miRNAs miR-26b-5p, miR-146a-5p, miR-93-3p, and miR-127-3p were progressively downregulated in serum before AKI diagnosis by serum creatinine after cardiac surgery [67]. Recently, the study showed that circulating miR-494 increased significantly in children with AKI after cardiopulmonary bypass [68]. Of note, emergent data suggested that the higher level of serum miR-210 and miR-494 in patients with sepsis-induced AKI were also correlated with poor prognosis and survival [69]. Additionally, a similar study was performed in CIN. The serum levels of three miR-30 family members (miR-30a, miR-30c, and miR-30e) showed more than a two-fold increase in comparison with those in patients who received contrast media but did not experience nephropathy [70]. Moreover, circulating miRNA-30a and miRNA-30e levels were also validated to increase in another CIN cohort study conducted by Sun et al. [71]. They found that circulating miR-188 could be a predictor for CIN [71].

### 5.2. Urinary miRNAs

Although many researchers study circulating miRNAs, our group focuses on urinary miRNAs for early identification of AKI development. First, given the diversity of AKI pathogenesis, the circulating miRNA might be much more complicated than urinary miRNA, especially during the complex immunologic reaction in sepsis-related AKI. Second, it is believed that renal tubular injury is involved in most kinds of AKI. Urinary miRNA would be much more specific under this condition. Of note, in the following studies about possible candidate miRNAs for AKI, their expression was often decreased compared to that in the respective control group. This phenomenon might raise questions about sample preservation or the accuracy of the detection method for further application in clinical use. After our first finding of the increase in urinary miR-494 in AKI patients, Ramachandran et al. further found that urinary miR-21, miR-200c, and miR-423 increased in AKI patients during ICU admission [72]. Likewise, urinary miR-30c-5p and miR-192-5p were significantly increased as early as 2 h after cardiac surgery in AKI patients [73]. Furthermore, urinary miR-30c-5p had better diagnostic value in postcardiac surgery-related AKI compared with protein-based markers such as neutrophil gelatinase-associated lipocalin (NGAL) and kidney injury molecule-1 (Kim-1) [73]. Therefore, urinary miRNA might be a much more promising potential biomarker than circulating miRNA for further translation into clinical use for the early detection of AKI due to its rapid increase after cellular injury.

## 6. miRNAs as a Potential Therapeutic Strategy in AKI

### 6.1. miRNA Targeted Therapy

miRNAs are critical players in epigenetic regulation and thus may have potential therapeutic (or preventive) effects. In our previous study, not only the level of miR-494 but also the urinary level of miR-16 in AKI patients was significantly higher than those in patients without AKI during ICU admission [77]. In a “proof of concept” study, mice received an intrarenal pelvic injection of lentivirus containing miR-16 or antisense-miR-16 before I/R injury. Unfortunately, the urinary miR-16 level was only partially reduced by in vivo lentivirus-mediated antisense-miR-16 gene transfer into the kidneys, and the attenuation of I/R-induced renal dysfunction was also incomplete compared to scramble control [77]. However, Wilingseder et al. intravenously administered antisense oligonucleotide (ASO) in mice and successfully inhibited miR-182 in the mouse kidneys for up to 96 h [78]. Subsequently, they demonstrated that the highly selective inhibition of miR-182-5p by ASO injection improved kidney function and histology significantly after I/R injury in mice [78]. The authors suggested that this antisense technology may have the potential to change the practice of human kidney transplantation. Likewise, the miR-668 mimic has been reported to protect mice from ischemic AKI by suppressing mitochondrial protein 18 kDa (MTP18) [79].

### 6.2. Stem Cell-Derived miRNAs (Stem Cell Secretome)

In addition to the potential miRNA-based therapy for AKI by miRNA mimics or anti-miRNA oligonucleotides, we emphasize the clinical use of microvesicles or exosomes not only as a spectacular delivery system for miRNA but also for stem cell-derived miRNAs. EVs are membrane-bound vesicles released by cells for intercellular communication. EVs are subdivided into exosomes, microvesicles, and apoptotic bodies. miRNAs are particularly enriched in exosomes and microvesicles. Stefano Gatti et al. found that the administration of microvesicles isolated from mesenchymal stem cells (MSCs) could protect rats against I/R injury, but RNase-treated microvesicles lost their protective effect [80]. Cantaluppi et al. also found that the injection of endothelial progenitor cell (EPC)-derived microvesicles into mice could protect the mouse kidneys from I/R injury but fibroblast-derived microvesicles could not. Of note, the EPC-derived microvesicles lost their renoprotective effect after treatment with RNase, nonspecific miRNA depletion by Dicer knockdown, or specific depletion of miR-126 and miR-296 by transfection of progenitor cells with miR-antagomirs [81]. Vinas et al. further studied human cord blood endothelial colony-forming cell (ECFC)-derived exosomes, which were enriched for miR-486-5p. Infusion of ECFC exosomes ameliorated I/R injury in vivo, which was associated with increased kidney miR-486-5p levels, PTEN levels, and increased phosphorylated Akt levels [82]. Similarly, exosomes from ECFCs transfected with anti-miR-486-5p lost their renoprotective effect in vivo and in vitro [82]. Recently, Zhu et al. also demonstrated that human-bone-marrow-derived mesenchymal stem cells exosomes protected against I/R injury in vivo and in vitro by delivering miR-199a-3p renal tubule cells [83]. These results suggest that miRNA-containing microvesicles or exosomes from stem cells, known as the stem cell secretome, utilize cell–cell communication and might be a novel potential therapeutic strategy for AKI.

## 7. Conclusions

The central dogma of molecular biology, where genetic material is transcribed into RNA and then translated into protein, provides a great starting point for our understanding of the principles of nature. The picture has been revised in light of emerging novel roles for noncoding RNA, especially miRNA. miRNAs may fine-tune large genetic networks or predominantly control specific main targets, thus playing critical roles in almost all biological cell functions. The accumulated evidence suggests that miRNAs play a vital role in the development and progression of AKI. There is no doubt that the more we understand the basic biology of miRNAs, the more we could apply them for the treatment of AKI patients. As our previous finding demonstrated, we would like to emphasize the role of urine miRNA in AKI (and most kidney diseases). There is still little known about the fundamental biology of urine miRNAs. For example, while most circulating miRNAs are bound solely by AGO proteins, but not encapsulated by microvesicles, whether most urine miRNAs share the same characteristic is still unclear.

Additionally, whether the stabilization of urinary miRNAs is associated with microvesicles or AGO2 (or both) is not completely understood, although a previous study showed that urinary miRNAs are resistant to trypsin [84] but not proteinase K [85]. Nevertheless, our unpublished data showed that the total amount of urine miRNAs was obviously increased in urothelial carcinoma patients followed by septic AKI patients and CIN patients compared to that in healthy volunteers. The delivery system of urine miRNAs might be different under different conditions, such as diseased vs. healthy state or tumorigenesis vs. inflammation. Further research in this area will improve our understanding of the physiological role and pathogenesis of urine miRNAs, and their mechanism of cell–cell communication.

In this article, we illustrated the possible model of intracellular and extracellular miRNAs in renal tubular cells after acute kidney injury to provide further research directions and innovative translation applications in Figure 1. In addition, in recent years, the growing preclinical data show that urinary miRNAs could serve as biomarkers for the early diagnosis of AKI. Favorably, our recent study demonstrated that the intrarenal administration of EVs enriched in ATF3 RNA exhibited a protective effect in a mouse model of I/R-induced acute renal injury. Bruno et al. further demonstrated that miRNAs in stem and progenitor cell-derived EVs might be a promising early intervention for AKI. Recently, Michaels et al. employed CRISPR/Cas9 to integrate miRNA silencing-mediated fine-tuners (miSFITs) into the 3′ UTR of a critical tumor suppressor gene to demonstrate genetically encoded fine-tuning of endogenous gene expression levels in mammalian cells [86]. These studies might turn a new page for the management of AKI, as AKI remains one of the leading causes of morbidity and mortality in hospitalized patients worldwide.

## Figures and Tables

**Figure 1 ijms-21-06738-f001:**
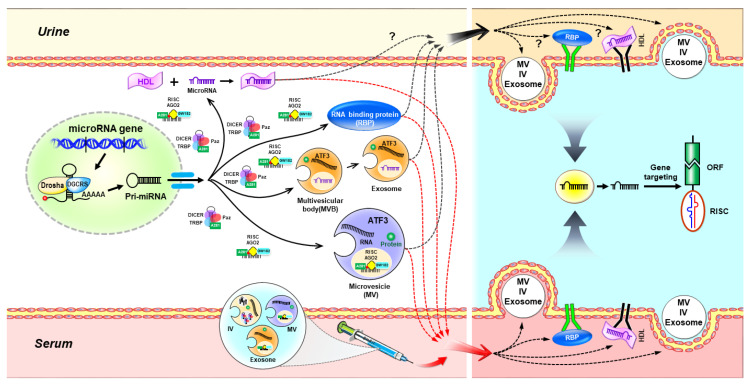
The possible model of intracellular and extracellular miRNAs after acute kidney injury in renal tubular cells. To respond to the acute injury, miRNAs are first transcribed as primary miRNAs (Pri-miRNAs) in renal tubular cells. Next, pri-miRNAs further cleave to the pre-miRNAs by the Drosha/DGCR8 microprocessor complex and then exported into the cytoplasm. In the cytoplasm, pre-miRNAs are then unwound by Dicer to the formation of double-stranded miRNAs which complex with argonaute (AGO) proteins. The mature miRNA guide strand is stably loaded into the RNA-induced silencing complex (RISC) along with AGO2 protein and GW182 to form mature RISC (miRISC) complex, while the other strand is thought to be degraded. The mature miRNAs in the miRISC can base pair with its target mRNAs and lead to the repression of protein translation or mRNA degradation. If the miRISC does not combine with its target strands, miRNAs can then be selected and sorted into the microvesicles (MVs), intermediate vesicles (IVs), or multivesicular bodies (MVBs). Exosomes are then secreted after translocation of MVBs from the cytoplasm to the plasma membrane by exocytosis. miRNAs could also secrete with the collaboration of high-density lipoprotein (HDL), AGO2, or other RNA-binding protein (RBP). Red arrows indicate the possible transport pathways of serum miRNAs (also named “circulating microRNAs”). Black arrows indicate the possible transport pathways of urinary miRNAs. Although most of these in urine remains unknown, the growing emergent data suggested urinary miRNAs could be novel biomarkers for acute kidney injury (AKI). Once miRNAs enter to the downstream target renal tubular cells, and then, again, form miRISC to base pair with its target mRNAs leading to mRNA silencing. Of note, the injection of miRNA targeted therapy, or stem-cell-derived miRNAs might be an innovative therapeutic strategy for AKI. ORF: open reading frame.

**Table 1 ijms-21-06738-t001:** MicroRNAs in acute kidney injury as novel biomarkers.

Possible Pathophysiology	Species	Serum	Urine	Ref.
I/R	Human, mice	-	↑ miR-494	[55]
I/R	Human	↑ miR-494	-	[68]
I/R, inflammation	Human	↓ miR-16, miR320 ↑ miR 210	-	[66]
I/R, inflammation	Human	↓ miR-101-3p, miR-127-3p, miR-210-3p, miR-126-3p, miR-26b-5p, miR-29a-3p, miR-146a-5p, miR-27a-3p, miR-93-3p and miR-10a-5p	-	[67]
I/R	Human	↑ miR-21	↑ miR-21	[63,65]
I/R	Human	↓ miR-21	↓ miR-21	[64]
I/R, inflammation	Human	-	↑ miR-21	[74]
I/R, inflammation	Human	↑ miR-21-3p	-	[75]
I/R	Human	↑ miR-192	-	[76]
I/R, inflammation	Human	-	↑ miR-21, miR-200c, and miR-423 ↓ miR-4640	[72]
I/R	Human	-	↑ miR-30c-5p, miR-192-5p	[73]
I/R, inflammation	Human, mice	-	↑ miR-16	[77]
CIN	Human	↑ miR-30a, miR-30c, and miR-30e	-	[70]
CIN	Human	↑ miR-30a, miR-30e, and miR-188	-	[71]

The upwards arrow represents the level of microRNA increases in AKI status, while the downwards arrow represents the level of microRNA decreases in AKI status.

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
