# Peer review of "MicroRNAs as Biomarkers and Therapeutic Targets in Inflammation- and Ischemia-Reperfusion-Related Acute Renal Injury"

_ijms, 2020, doi:10.3390/ijms21186738_

Round 1

Reviewer 1 Report

An interesting review, written in an accessible and understandable manner, in my opinion, is suitable for publication in the International Journal of Molecular Sciences. The only remark concerns the updating of the literature - the review contains only 3 citations from 2019, the lack of data from 2020.

Author Response

We are very grateful for the helpful and perceptive comments from the editorial office and the two referees. We have carefully reviewed the suggestions, and our responses are as follows.

Point 1: An interesting review, written in an accessible and understandable manner, in my opinion, is suitable for publication in the International Journal of Molecular Sciences. The only remark concerns the updating of the literature - the review contains only 3 citations from 2019, the lack of data from 2020.

Response 1: We rediscovered the associated literature of the latest two years and added those new data to our manuscript from ref. 68,69, 83 (2019) and 62, 75 (2020) in line 310-312, 312-313, 372-374, 286-288 and table 1.

Reviewer 2 Report

This manuscript is a review  on a very hot topic, namely the role of micro RNA (miRNA) in the pathogenesis of acute kidney injury (AKI) due to inflammation or to an ischemia/reperfusion insult and their potential usefulness as  biomarkers and/or as therapeutical targets.

The Authors performed an accurate and deep overview  of the  most relevant publications  in a very clear and complete way.

I have only minor  suggestions for the Authors:

  • Page 2, Definitions of AKI: the authors could also comment on the fact that definition criteria of AKI based only on the increase, even if minimal, of creatinine do not intercept early kidney damage that precedes the reduction of glomerular filtration rate
  • although the authors explicitly state that they only deal with the forms of AKI related to inflammation and ischemia-reperfusion, probably a table with the other causes of AKI in which some role of miRNAs has been suggested in the literature could give a more complete information
  • some likely typing errors should be corrected: e.g., pg 6, line 276, “ is the data are difficult….” ?; pg 7,  line 318, the first Author of the reference n. 69 is quoted  in the text by his first and not family name; pg 9, line 391, “ cell-derived EVs might a promising…” missing “be” ?

Author Response

We are very grateful for the helpful and perceptive comments from the editorial office and the two referees. We have carefully reviewed the suggestions, and our responses are as follows.

Comments and Suggestions for Authors

I have only minor suggestions for the Authors:

Point 1: Page 2, Definitions of AKI: the authors could also comment on the fact that definition criteria of AKI based only on the increase, even if minimal, of creatinine do not intercept early kidney damage that precedes the reduction of glomerular filtration rate

Response 1: We added a short paragraph to enhance the limitation of serum creatinine-based definition criteria of AKI as your constructive suggestion in line 57-62.

Point 2: although the authors explicitly state that they only deal with the forms of AKI related to inflammation and ischemia-reperfusion, probably a table with the other causes of AKI in which some role of miRNAs has been suggested in the literature could give a more complete information

Response 2: We added a new table that listed miRNAs have been suggested as novel biomarkers in the current human studies.

Point 3: some likely typing errors should be corrected: 

e.g., 

pg 6, line 276, “ is the data are difficult….” ?; 

pg 7,  line 318, the first Author of the reference n. 69 is quoted  in the text by his first and not family name; 

pg 9, line 391, “ cell-derived EVs might a promising…” missing “be” 

Response 3: The manuscript was carefully edited for those typing errors and grammar correction again.

Page 2, line 45-46, “gene RNexpression” change to “gene RNA expression.”

Page 3, line 131, “in most condition” change to “in most conditions.”

Page 4, line 147, “Another example are” change to “Another example is”

Page 6, line 276,” administration of exosomal Flag” change to “the administration of exosomal Flag.”

Page 7, line 281, “ is the data are difficult….” change to ”the data are difficult….”

Page 7, line 329,” Krithika et al. change to “Ramachandran et al.”

Page 10, line 406, “ cell-derived EVs might a promising…” change to “cell-derived EVs might be a promising.”

Page 10, line 407, “Yale et al.” change to “Michaels et al.”